# Predicting locations of cryptic pockets from single protein structures using the PocketMiner graph neural network

Artur Meller [1,2,5], Michael Ward[1,5], Jonathan Borowsky[1], Meghana Kshirsagar[3], Jeffrey M. Lotthammer [1], Felipe Oviedo[3], Juan Lavista Ferres [3] & Gregory R. Bowman [1,4] ✉

Cryptic pockets expand the scope of drug discovery by enabling targeting of proteins currently considered undruggable because they lack pockets in their ground state structures. However, identifying cryptic pockets is labor-intensive and slow. The ability to accurately and rapidly predict if and where cryptic pockets are likely to form from a structure would greatly accelerate the search for druggable pockets. Here, we present PocketMiner, a graph neural network trained to predict where pockets are likely to open in molecular dynamics simulations. Applying PocketMiner to single structures from a newly curated dataset of 39 experimentally confirmed cryptic pockets demonstrates that it accurately identifies cryptic pockets (ROC-AUC: 0.87) >1,000-fold faster than existing methods. We apply PocketMiner across the human proteome and show that predicted pockets open in simulations, suggesting that over half of proteins thought to lack pockets based on available structures likely contain cryptic pockets, vastly expanding the potentially druggable proteome.

Protein structural fluctuations often lead to the formation of cryptic pockets[1–5], which present druggable sites beyond pockets apparent in experimentally-determined structures. From a drug development perspective, targeting these cryptic pockets provides a number of compelling opportunities. For example, proteins that lack an obvious pocket in the native, folded structure may appear undruggable, but could be targeted via cryptic pockets. Additionally, while molecules that target an orthosteric site are obligate inhibitors, molecules that target a cryptic pocket can modulate protein function via inhibition or activation[6,7]. Finally, while orthosteric sites are often highly conserved across proteins that need to bind the same ligand, cryptic pockets are likely less conserved[8,9]. This opens the possibility of developing molecules that have improved specificity.

While cryptic pockets are alluring drug targets, it remains challenging to find and target them intentionally. Most known cryptic pockets were discovered serendipitously by screening for inhibitors and solving structures for hits[2,4,10]. While this process has unveiled cryptic pockets, it does not specifically select for compounds that target cryptic pockets and is both costly and labor-intensive. More-over, the discovery of cryptic pockets through this approach is rare because the lack of a priori knowledge of the cryptic pocket structure prohibits the design of a small molecule library targeting the pocket. Molecular dynamics simulations are another means to identify cryptic pockets. Simulations provide an atomically-detailed ensemble of structures that a protein adopts in solution, which commonly reveals cryptic pockets that can be used as a template for drug design[11–16]. However, molecular dynamics simulations are computationally expensive, making it infeasible to screen large numbers of targets for cryptic pockets.

Because identifying cryptic pockets is resource-intensive, a screening method that quickly indicates if a protein is likely to have any

[1]Department of Biochemistry and Molecular Biophysics, Washington University in St. Louis, 660 S. Euclid Ave., Box 8231, St. Louis, MO 63110, USA. [2]Medical Scientist Training Program, Washington University in St. Louis, 660 S. Euclid Ave., St. Louis, MO 63110, USA. [3]AI for Good Research Lab, Microsoft, Redmond, WA, USA. [4]Department of Biochemistry and Molecular Biophysics, University of Pennsylvania, 3620 Hamilton Walk, Philadelphia, PA 19104, USA. [5]These authors contributed equally: Artur Meller, Michael Ward. ✉e-mail: grbowman@seas.upenn.edu

cryptic pockets would be extremely valuable. During the SARS-CoV-2 pandemic, researchers rapidly solved numerous experimental structures of different viral proteins[17]. Similarly, recent advances in protein structure prediction have made many protein structures available for structure-based drug design[18–20]. Proteins with pockets in their ground state experimental structures may be prioritized as drug targets. However, in cases where proteins lack ground state pockets or where the design of specific modulators is challenging, an algorithm designed to predict which proteins have cryptic pockets would also be useful for prioritizing which proteins to target.

CryptoSite is an outstanding example of a supervised machine learning algorithm that takes a protein structure as input and predicts ligand-binding cryptic pockets[21]. Briefly, CryptoSite is trained to identify amino acid residues that will transition from an orientation that is incompatible with ligand binding to an orientation that has been verified to accommodate a ligand based on a set of 84 confirmed cryptic pockets derived from the Protein Databank (PDB). Even with this relatively small dataset, CryptoSite achieves good accuracy classifying whether an amino acid residue will participate in a cryptic pocket (ROC-AUC = 0.83). Achieving this performance requires ~1 day to run for a single protein because one of CryptoSite's input features is simulation data, which it needs to generate on-the-fly for each prediction. The algorithm takes a performance hit without using simulation data as a feature (ROC-AUC = 0.74). A faster algorithm that achieves the same or better accuracy would be of tremendous value for prioritizing potential drug targets based on their likelihood of having useful cryptic pockets.

We hypothesized that an algorithm trained to predict the probability a pocket forms within a fixed amount of simulation time would accurately identify cryptic pockets in ligand-free experimental structures. Specifically, we propose using simulations to evaluate if each residue in a protein structure can rearrange its orientation to participate in a cryptic pocket as part of its thermal fluctuations. In contrast to CryptoSite, which relies on a small number of examples where a ligand is known to bind in a cryptic site, the proposed training scheme does not necessitate examples of ligands bound in cryptic pockets. Instead, models can be trained on structural ensembles (e.g., from molecular simulations) that contain examples of pocket opening events. One benefit of the proposed training scheme is that at least an order of magnitude more training examples can be obtained to train models (e.g., thousands of cryptic pocket opening events can be obtained from simulations).

In the current study, we trained a graph neural network to predict where pockets are likely to open in molecular dynamics simulations (Fig. 1) and then tested whether it can predict the locations of cryptic pockets from single, experimentally derived structures (Supplementary Fig. 1). Specifically, we trained a model to take a structure of a protein and forecast whether each residue will participate in the formation of a cryptic pocket over the course of a short simulation initiated from that structure. Our training dataset consists of a previous molecular dynamics simulations study that identified cryptic pockets across most proteins in the SARS-CoV-2 proteome[12], simulations from a previous study of Ebolavirus VP35[5], and a set of 16 proteins with known ligand-binding cryptic pockets that we simulated for this study. To test the ability of PocketMiner to predict the locations of ligand-binding cryptic pockets from single structures, we curated a dataset of 39 examples of cryptic pockets from the PDB. For each of these systems, there is a structure of the *apo* protein where the cryptic pocket is absent and a *holo* structure with a ligand bound in the cryptic pocket. We analyzed this dataset to learn a taxonomy of cryptic pockets and tested PocketMiner's ability to identify the different types of cryptic pockets. Finally, we applied PocketMiner to the entire human proteome and highlighted new cryptic pockets we propose to be promising drug target candidates.

## Results

### Known cryptic pockets open rapidly in simulations

There have been few studies which systematically evaluate whether molecular dynamics (MD) simulations recapitulate known cryptic pockets. In previous work, MD simulations captured opening at known cryptic sites in TEM β-lactamase[22], Interleukin-2[23], and several other protein drug targets[13,24]. MD simulations have also identified novel cryptic pockets that were experimentally confirmed in thiol labeling experiments[5,25]. However, these studies focused on a narrow set of proteins or did not evaluate whether pocket formation was localized to known ligand-binding sites. Given the recent interest in using MD simulations for cryptic pocket discovery, there is a need for a systematic evaluation of how well MD simulations recapitulate known cryptic pockets. Such a study could dually generate data to train a machine learning model to identify the locations of cryptic pocket formation.

Hence, we conducted unbiased adaptive sampling MD simulations of 16 proteins known to form cryptic pockets from *apo*, or ligand-free, starting structures. Eleven of the pairs have ligand-binding residues which were closer together in the *apo* structure than in the *holo* structure (i.e., they required an opening motion to form the pocket). Multiple different types of motion are represented, including three cases of secondary structure change. For each protein, we ran 2 μs of adaptive sampling simulations using the Fluctuation Amplification of Specific Traits (FAST) algorithm[26]. Specifically, we launched 10 parallel simulations from the *apo* structure, then constructed a Markov state model (MSM)[27,28] of the conformational ensemble and prioritized structures for the next round of simulations using a ranking function that balances exploitation (i.e., prioritizing states with large pockets) with exploration. This procedure was repeated four times to generate 5 'swarms' of simulations, each consisting of 400 ns of aggregate simulation (10 simulations 40 ns in length).

To our surprise, we find that the large majority of cryptic pockets open in just 10 parallel simulations of 40 nanoseconds (Fig. 2). Pockets were considered open if the pocket volume of a simulated structure reached or exceeded the *holo* crystal structure pocket volume (see Methods). One of the proteins had a cryptic volume that was larger in *apo* than in *holo* and was thus eliminated from further analysis. Thirteen out of the remaining 15 proteins open in the first 10 simulations of 40 nanoseconds. One other, elongation factor TU, opens after additional rounds of adaptive sampling (five rounds of 10 parallel 40 ns simulations). Only one protein, Niemann-Pick C2 Protein, remains closed even after the 5 rounds of adaptive sampling simulations. We hypothesize that cryptic pocket opening is not observed for this protein because the ligand that binds in the cryptic pocket is highly hydrophobic[29]. Encouragingly, we also find that cryptic pockets that form in simulation are mostly localized to known ligand-binding cryptic sites with the largest pocket volume changes occurring at ligand-binding sites (Supplementary Fig. 2, Supplementary Table 6). Altogether, the rapid pocket opening we observe suggests that for most smaller proteins a modest amount of simulation data may be enough to discover cryptic pockets. Additionally, this finding suggests that machine learning models trained to predict cryptic pocket formation over small simulation time windows (i.e., 40 ns) would also be able to identify cryptic sites in ligand-free experimental structures.

### Graph neural networks accurately predict the time evolution of pockets in simulation

Given a starting structure, we reasoned that one could predict where cryptic pockets will form in that structure during an MD simulation. Though MD simulations are stochastic, and no two simulations launched from the same structure will follow the exact same time evolution, we reasoned that sufficiently long simulations would sample similar pocket opening events. For example, a residue that is loosely packed with few specific interactions with its neighbors might move to

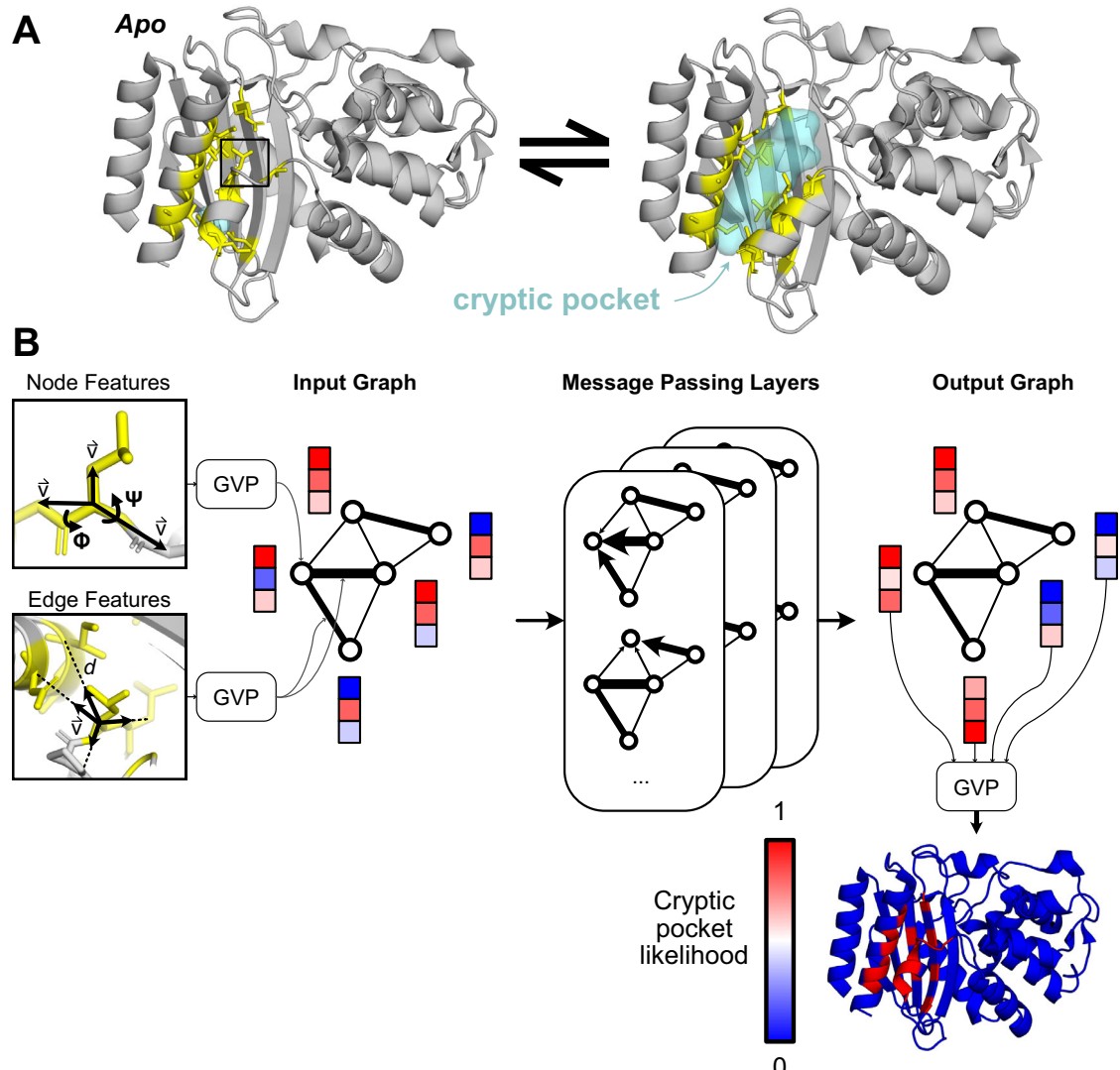

**Fig. 1 | PocketMiner uses graph neural networks to predict cryptic pocket formation. A** Proteins exist in an equilibrium between different structures, including experimentally derived structures that lack cryptic pockets (left, PDB ID 1JWP)[71] and those with open cryptic pockets (right, PDB ID 1PZO)[4]. Residues lining the cryptic pocket are shown in yellow sticks. **B** PocketMiner relies on a series of message passing layers to exchange information between residues and to generate encodings that can predict sites of cryptic pocket formation. On the left, we show the structural features that are fed into an input graph following transformations by Geometric Vector Perceptron (GVP) layers. Node features include backbone dihedral angles as well as forward and reverse unit vectors (for a full list see Methods). Edge features include a radial basis encoding of the distance between residues and a unit vector between their alpha-carbons. In the middle, we show how the input graph is transformed by messaging passing layers which influence a residue's embedding based on its neighbors' node embeddings as well as its edge embeddings. On the right, we show that the node embeddings from the output graph are used to make predictions of cryptic pocket likelihood following another GVP transformation. Finally, at the bottom right, we show an idealized prediction for the protein shown in A.

create a pocket consistently across multiple simulations. Conversely, a residue that is tightly packed with strong interactions with its neighbors would be more likely to remain locked in its initial position across independent simulations.

To confirm our intuition, we evaluated the consistency of pocket openings across independent simulations of known cryptic pocket openings. We calculated pocket volumes using the LIGSITE algorithm[30] for each structure visited in a simulation (2000 structures per 40 ns simulation) and assigned pocket volumes to nearby residues. Specifically, we determined how many LIGSITE pocket grid points are within 5 Å of each residue. We then binarized residues into those that were part of a pocket opening event and those that did not form pockets. A residue was considered a positive example if at any point in simulation the nearby pocket volume determined by the LIGSITE algorithm increased by more than 40 Å$^3$ relative to its assigned pocket volume in

the starting structure (see Methods). We find that independent 40 ns simulations launched from the same starting structure produced similar labels, with the vast majority of residues either always opening or always remaining closed (Supplementary Fig. 3). When we visually inspect a protein with a known cryptic pocket opening like TEM 1 β-lactamase, we observe opening in similar hotspots across simulations (Supplementary Fig. 2).

Given that independent simulations converge to similar labels, we set out to develop an algorithm that predicts sites of cryptic pocket opening events in simulation. Specifically, we combined our simulation data of known cryptic pocket openings with additional adaptive sampling simulations of SARS-CoV-2 proteins[12]. This dataset included 37 proteins and 2400 independent MD simulations at least 40 ns in length. We generated labels for each residue in each 40 ns window's starting structure based on whether that residue participates in a

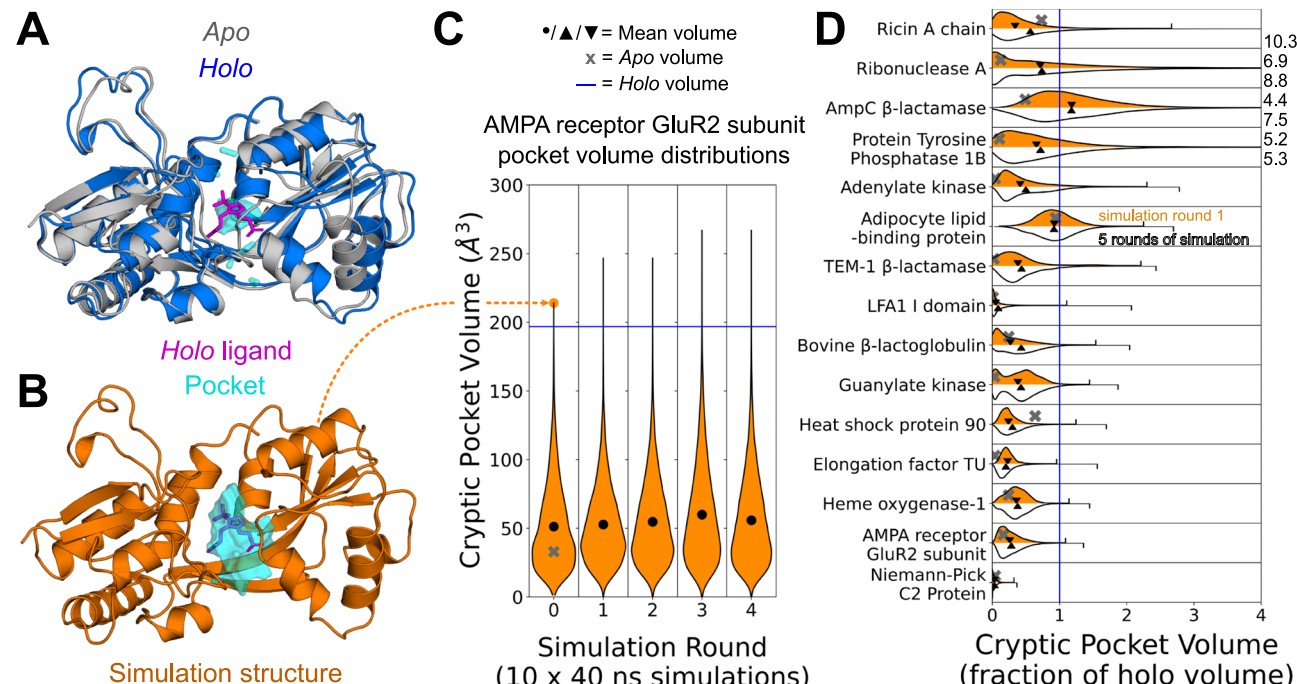

**Fig. 2 | Cryptic pockets rapidly open in simulations started from closed *apo* structures. A** Structural overlay of the GluR2 subunit of the AMPA receptor protein in its *apo* (grey, PDB: 1MY0[72]) and *holo* (blue, PDB: 1N0T[73]) conformations reveals that a loop and helix must shift to form a cryptic pocket. **B** A structure from an MD simulation started from the closed *apo* conformation with a pocket at the binding site with a volume exceeding that of the *holo* binding site pocket (orange). **C** MSM-weighted violin plots of cryptic pocket volume in simulations of the GluR2 subunit of the AMPA receptor show that the ligand binding site exceeds the volume seen in the *holo* conformation within only 10 parallel simulations of 40 ns each. Cryptic pocket volume was calculated by assigning each LIGSITE[30] grid point to the single nearest residue and summing up the volumes assigned to all cryptic pocket residues (see Methods). The grey 'X' indicates the *apo* binding site cryptic pocket volume, the blue line indicates the holo volume, and the black dot indicates the MSM-weighted mean volume of all frames for each round of simulation. **D** MSM-weighed distributions of cryptic pocket volumes from simulation of 15 different proteins show that even 10 short simulations are usually sufficient to reach the cryptic pocket volume of the *holo* structure. The pocket volume distribution as a fraction of the *holo* volume for 10 short simulations (400 ns of cumulative simulation) is shown in orange while the distribution for five rounds of adaptive sampling (2 microseconds of sampling) is shown in white. The grey 'X' indicates the *apo* cryptic pocket volume as a fraction of *holo*, and the downward-pointing and upward-pointing triangles represent the MSM-weighted mean volume after 10 simulations (1 round) or 5 rounds of adaptive sampling respectively. The maximum cryptic pocket volumes as a fraction of *holo* reached in simulation are shown with tick marks or given numerically to the right of the plot if they lie outside the plot's range. Source data are provided as a Source Data file.

cryptic pocket at any point in the next 40 ns of simulation (Fig. 3A). Altogether, this dataset included 941,650 unique examples. We then used this data to train a graph neural network based on the geometric vector perceptron graph neural network (GVP-GNN)[31] as well as a 3D convolutional neural network (3D-CNN)[32]. Both these architectures have demonstrated robust performance on other protein structure prediction tasks.

We find that both the GVP-GNN and the 3D-CNN learn to accurately classify whether a given residue will form a cryptic pocket in simulation based on a starting structure. We split the 37 protein simulation datasets into 5 folds by protein and used 5-fold cross-validation to measure the performance of both architecture types. For each split, we separately assessed how choices in model hyperparameters (e.g., dropout rate) and training set up (e.g., class balancing scheme) affected performance on 1 validation fold (3 folds were used for training). The GVP-GNN and 3D-CNN that performed best on each validation fold (Supplementary Table 2-4) for a certain split was then assessed using a held-out test fold (Supplementary Table 1). Across the 5 splits, the best GVP-GNN model achieves an average test PR-AUC of $0.44 \pm 0.12$ (average ROC-AUC: $0.83 \pm 0.04$; Fig. 4B–C). The 3D-CNN performs similarly (PR-AUC: $0.41 \pm 0.05$; ROC-AUC: $0.79 \pm 0.02$). At high levels of recall (0.6-0.8), the GVP-GNN makes fewer false positive predictions, as demonstrated by the higher precision on the right side of the precision-recall curve (Fig. 4B). As a result, we decided to proceed with the GVP-GNN architecture for further prediction tasks. Overall, these findings suggest that it may be possible to identify sites

where cryptic pockets form without computing intermediate states (e.g., with MD simulations) when given a structure of the native, folded state of a protein.

## Cryptic pocket dataset reveals forward and reverse motions

To evaluate if our models could detect sites of cryptic pocket formation from experimentally validated ligand-binding cryptic pockets, we first curated a dataset of cryptic pockets. Cimermancic et. al.[21] previously identified 93 *apo-holo* protein structure pairs containing cryptic pockets, which will be referred to here as the CryptoSite set. While useful, the CryptoSite set contains relatively few proteins in which large conformational changes are necessary for pocket formation[33] (Supplementary Fig. 4). Furthermore, the PDB has roughly doubled in size since the version used to generate the CryptoSite set was downloaded, and Sun et. al. observed that some proteins in the CryptoSite set had additional *apo* structures in which the pocket was open[34]. We reasoned that curating a novel cryptic pocket dataset would improve our understanding of cryptic pockets and provide a test set for our model that has increased coverage of the large and diverse space of cryptic pocket structures and motions. Thus, we filtered the Protein Data Bank (PDB) to identify 38 *apo-holo* protein structure pairs containing 39 cryptic pockets with large root mean square deviations between *apo* and *holo* (see Methods, Supplementary Fig. 4).

The resulting collection of cryptic pockets, called the PocketMiner dataset, includes pockets formed by multiple types of conformational

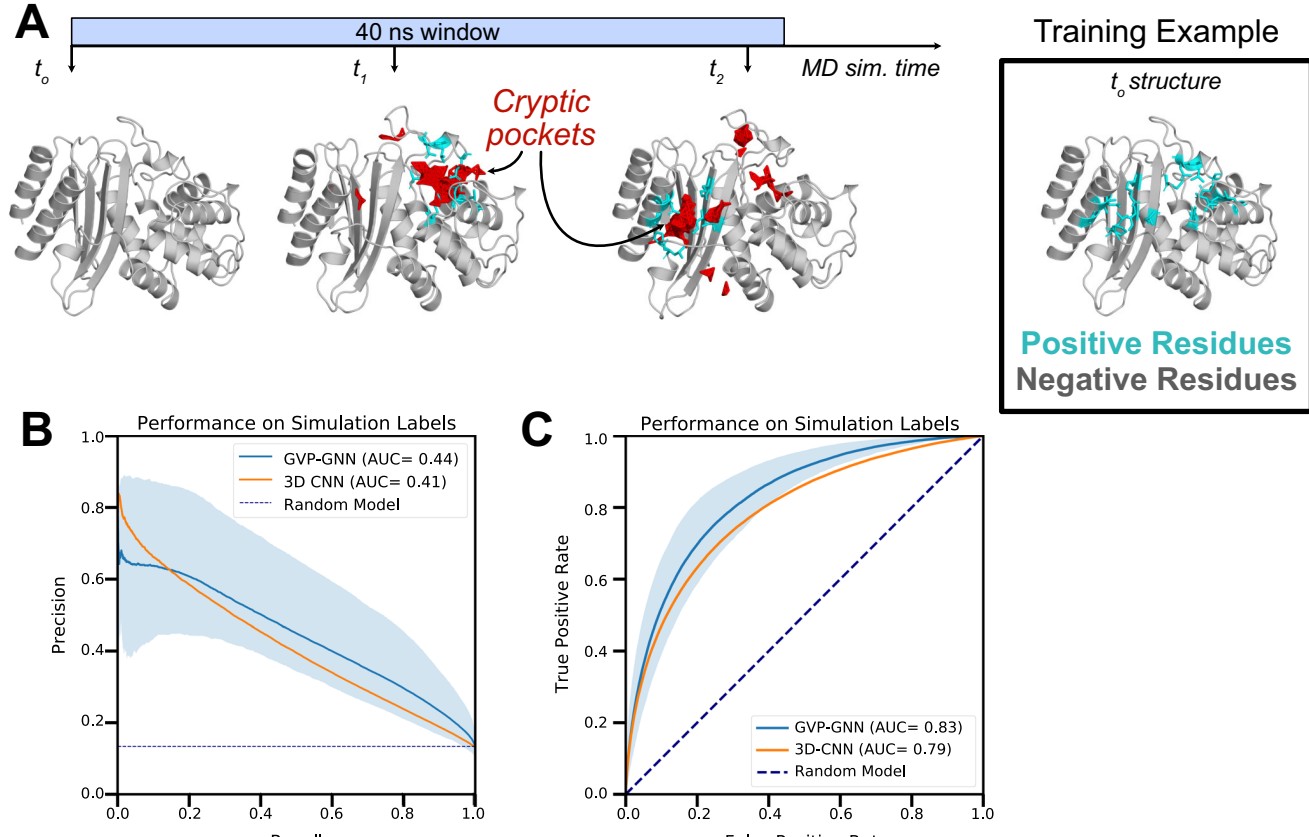

**Fig. 3 | Graph neural networks accurately forecast the sites of pocket formation in new simulations. A** We used MD simulations to generate training labels by tracking where cryptic pockets (shown in red) form. Residues were labeled as positive examples if a cryptic pocket formed nearby that residue at any point in simulation (shown in cyan). As an illustration, we show the TEM β-lactamase protein, which forms cryptic pockets at two separate sites (i.e., the horn pocket and the omega loop pocket) in a single MD simulation. Each opening event marks the nearby residues as positive examples, so the training labels for this window reflect both opening events with residues around both pockets marked as positive examples. The starting structure can then be fed to a graph neural network that predicts where cryptic pockets will form. **B** Precision-recall curves across 5 different folds demonstrate that a graph neural network trained with simulation starting structures and labels derived from simulation intermediates predicts which residues will form cryptic pockets in simulation (mean PR-AUC of 0.44). The shaded region represents variation in performance across folds with the top of the shaded region tracing the curve with the highest PR-AUC and the bottom of the shaded region tracing the curve with the lowest PR-AUC. GVP-GNN refers to the Geometric Vector Perceptron-based Graph Neural Network; 3D-CNN refers to the 3D Convolutional Neural Network. **C** Receiver operating curves across 5 different folds demonstrate robust performance (mean ROC-AUC of 0.83). Shading represents variation across folds (worst to best performance across 5 folds). Source data are provided as a Source Data file.

changes. Interestingly, we notice that many pockets form via closing motions, rather than the canonical pocket opening motion. For these "reverse" pockets, structural elements start far apart, such that there is little or no pocket present in the *apo* structure, but come together to form a lid or walls, creating a cavity where a ligand can bind in *holo* (Fig. 4D). Most pockets are formed primarily by a single type of motion, although in several cases multiple types contribute significantly. Across forward and reverse pockets, we observe four common structural rearrangements. First, loops can move apart to create space for an incoming ligand or clamp down to create a wall or lid over a ligand (Fig. 4A). Second, secondary structural elements can shift via a translation and/or rotation (Fig. 4B). Third, secondary structural elements can change to, or form from, a loop (Fig. 4C). Fourth, interdomain motions can create space for ligands to bind (Fig. 4D). Thus, our cryptic pocket dataset captures a diverse set of conformational rearrangements and represents a challenging benchmark for evaluating machine learning models.

### PocketMiner accurately predicts ligand-binding cryptic pockets from ligand-free crystal structures

Given that the GVP-GNN accurately predicts where cryptic pockets will form in simulation, we wondered if this network architecture could be used to predict the sites of cryptic pocket formation in experimental structures. If known cryptic sites form quickly in simulations, then a GVP-GNN that learns to predict where pocket formation occurs in simulation might reasonably identify ligand-binding cryptic sites in experimental structures.

As a result, we took all proteins from our previous training set and generated training labels based on local changes in LIGSITE pocket volume and fpocket druggability scores[35] (see Methods). We decided to use fpocket labeling schemes since druggability scores consider not only the geometry of a pocket but also the chemical environment of a pocket. We trained GVP-GNNs using LIGSITE-derived labels only, fpocket-derived labels only, and a combination of both labels (i.e., where a network is trained for several epochs using one labeling scheme and then switches to learning from the other labels for several more epochs). Though previous studies have found that LIGSITE and fpocket can struggle to correctly rank which pockets bind ligands[36], our labels did not consider the ranking between pockets in a single structure. Instead, our labels were based on the change in the LIGSITE pocket volume or the maximum fpocket druggability score in the vicinity of a residue over the course of MD simulations. We then evaluated if models trained using these labels could distinguish residues at ligand-binding cryptic sites from residues that do not form cryptic pockets.

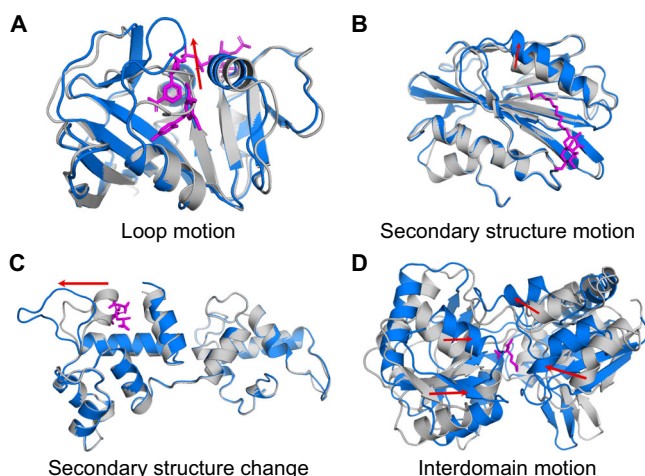

**Fig. 4 | PocketMiner's validation dataset contains diverse types of conformational changes that lead to cryptic pocket formation.** *Apo* structures are shown in gray, *holo* structures are shown in blue, and ligands are shown in magenta. The red arrows highlight the main conformational change occurring between *apo* and *holo*. **A** Cryptic pockets can arise as a result of loop motions like those seen in dihydrofolate reductase (apo PDB: 2W9T[74], holo PDB: 2W9S[74]); (**B**) secondary structure element motions such as those seen in lipoprotein LpqN (apo PDB: 6E5D[75], holo PDB: 6E5F[75]); (**C**) secondary structure changes like those observed in calcium- and integrin-binding protein 1 (apo PDB: 1Y1A chain A[76], holo PDB: 1Y1A chain B); (**D**) or interdomain motions as in nopaline-binding periplasmic protein (apo PDB: 4P0I[77], holo PDB: 5OTA[78]). We refer to examples like those shown in **A–C** as forward pockets because the residues adjacent to the ligand in the *holo* structure are farther from one another than they are in the *apo* structure. We refer to examples like that in D as reverse pockets because the residues adjacent to the ligand in *holo* are closer together than they are in *apo*.

To evaluate if our model can distinguish sites that do not form cryptic pockets, we curated a dataset of negative examples consisting of rigid proteins and other proteins that were the targets of extensive drug screening. In previous work, negative examples were defined as any residue not known to participate in a ligand-binding cryptic pocket[21]. However, many of these sites may have the potential to form ligand-binding cryptic pockets. To increase our confidence in our negative examples, we only used residues from proteins known to be extremely rigid or extremely stable or residues from proteins that have been put through extensive drug screens. Our dataset of hyper rigid proteins included three designed microproteins and a handful of proteins with unusually rigid folds or crystal-like properties (see Methods). We ran MD simulations of these proteins to verify that they do not form cryptic pockets and removed any residues that were near small pockets from the model evaluation dataset (see Methods). To supplement this relatively small number of negative examples for model evaluation, we also identified proteins that had been the subjects of extensive drug screens and pulled out residues where ligands did *not* bind (Supplementary Fig. 5). Our reasoning was that sites where ligands did *not* bind would be highly unlikely to form cryptic pockets given that these proteins had been co-crystallized or soaked with a large number of ligands. To build additional confidence in our negative labels, we also ran simulations of these proteins and removed any residues that were the sites of cryptic pocket opening in simulation (Supplementary Fig. 6). Thus, we assembled a collection of negative examples with both experimental and simulation evidence to support our labels.

We find that our final model, referred to as PocketMiner, achieves very good performance at discriminating residues that form cryptic pockets from those that do not (ROC AUC: 0.87). PocketMiner was trained for 20 epochs with LIGSITE-derived labels and refined for 1 epoch with labels derived from fpocket druggability scores. To test

PocketMiner, we made predictions on a total of 24 *apo* structures that form ligand-binding cryptic pockets, 4 hyper-rigid proteins, and 7 proteins that were the subjects of extensive ligand screening. All of the proteins in the test set had less than 55% sequence identity with proteins in the training set (Supplementary Fig. 7, Supplementary Table 5). In total, there were 563 residues that form cryptic pockets and 1283 residues that do not form cryptic pockets in our test set. Among *apo* structures that form cryptic pockets, our model predicts a high likelihood of cryptic pocket formation at the experimental cryptic site (Fig. 5A, Supplementary Fig. 7). We achieve robust performance across several classes of cryptic pocket opening (Fig. 5C, Supplementary Fig. 7, Supplementary Table 9). Conversely, our model predicts a low probability of cryptic pocket formation for all the hyper-rigid proteins and correctly assigns low predictions to regions that are unlikely to form cryptic pockets from proteins with extensive ligand binding (Fig. 5B, Supplementary Fig. 8). When we compared the performance of our model against CryptoSite, we find that the two methods give very similar performance, with PocketMiner providing a small advantage (ROC-AUC: 0.87 for PocketMiner vs. 0.85 for CryptoSite). In particular, we find that PocketMiner predicts fewer false positives among rigid proteins and sites that do not bind ligands, suggesting that PocketMiner may be a more useful screening tool (Supplementary Table 8). Moreover, we note that while CryptoSite must run simulations that often take several hours to generate a prediction, our model makes predictions in under a second (>1000x improvement in prediction time).

## PocketMiner predicts thousands of cryptic pockets across the human proteome

Given its ability to predict the locations of cryptic pockets accurately and rapidly, PocketMiner can be used to search for cryptic pockets across large numbers of protein structures. Thanks to decades of effort in structural biology and the high accuracy of protein structure prediction achieved by AlphaFold 2[19], there are proposed protein structures for all genes in a standard human genome. We hypothesized that PocketMiner could help uncover new opportunities for designing drugs against human proteins that lack clear drug binding pockets in their native, folded structures. Even in cases where there already exists a clear binding pocket, PocketMiner may identify cryptic pockets in other regions of the protein that could exert allosteric control over the protein's function or be used for the design of drugs with better specificity.

To identify cryptic pockets across the human proteome, we applied PocketMiner to over 10,000 human genes. For each gene, we prioritized using high resolution, experimentally determined structures, but used the AlphaFold predicted structure as an alternative when there were no available experimental structures. Finally, we discarded structures that had long stretches of predicted disorder (see Methods). We binned structures into three categories: proteins with ground state pockets, proteins with cryptic pockets only, and proteins which lack both. Each structure was binned based on calculating its largest LIGSITE pocket volume and its highest PocketMiner prediction (see Methods).

We find that several thousand proteins have predicted cryptic pockets despite lacking pockets in their ground state structures (29.4% of proteins in our set, Fig. 6A), expanding the fraction of proteins with single structured domains that are likely to form pockets to over 80%. A smaller number of single structured domain proteins lack both ground state and cryptic pockets (18.5%) according to our analysis. The large number of predicted cryptic pockets suggests that it may be fruitful to run a drug screen on a protein even if there are no obvious small molecule binding sites present in its native, folded structure. Ideally, one would use PocketMiner on some group of proteins of therapeutic interest, identify which ones are likely to have cryptic pockets, then simulate the structural dynamics of that protein

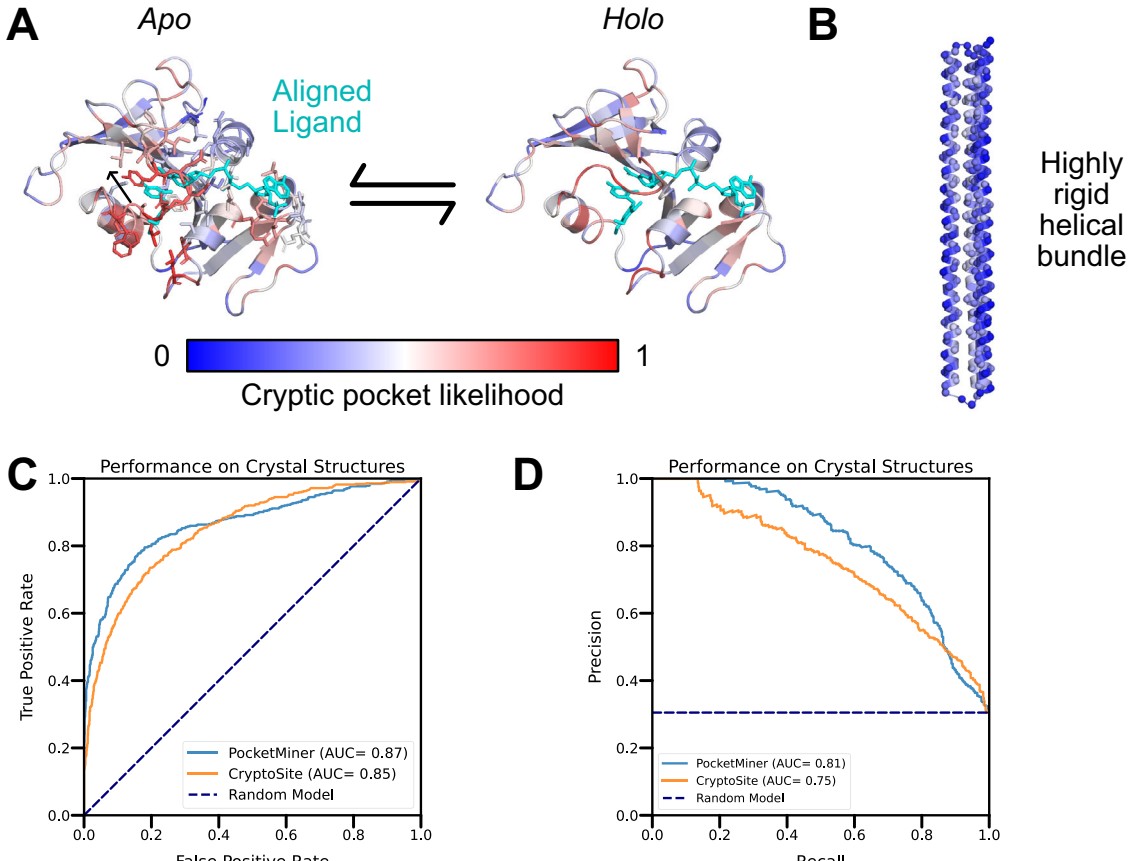

**Fig. 5 | The PocketMiner graph neural network accurately detects sites of cryptic pocket formation in experimental structures. A** PocketMiner predicts a high likelihood of cryptic pocket formation at the site of ligand binding. The ligand (cyan) from the aligned *holo* structure is shown on the *apo* structure to highlight the steric clash between the ligand and a loop that must move to create the *holo* binding site. Though PocketMiner only uses the *apo* structure to generate a prediction (blue indicates low probabilities of cryptic pocket formation while red indicates high probabilities), the predicted labels are also shown on the *holo* structure to highlight that those high predicted labels cluster near the ligand binding site. **B** PocketMiner correctly predicts that the probability of pocket formation is low for a highly rigid helical bundle (PDB: 4TQL[79]) that did not form large cryptic pockets in simulation. **C** Receiver Operating Curve for residue-level cryptic site detection shows that PocketMiner achieves a better performance than CryptoSite despite running >1000x faster. **D** A precision-recall curve highlights that at high levels of recall (0.6–0.8) PocketMiner predicts fewer false positives. Source data are provided as a Source Data file.

to identify druggable structures that are adopted in its structural ensemble. PocketMiner's residue-level pocket site predictions could inform the use of adaptive or enhanced sampling methods targeting the region of a protein predicted to contain a cryptic pocket.

To demonstrate the utility of PocketMiner, we used it to help identify cryptic pockets in proteins involved in the Jak/Stat pathway implicated in several cancers[37]. First, we applied PocketMiner to all proteins in the KEGG[38] pathway called "Pathways in Cancer" (KEGG ID: 05200). We identified *PIM2* as having a high cryptic pocket prediction. *PIM2* is a serine/threonine kinase proto-oncogene with a known orthosteric ligand-binding site that has been the target of drug screens[39]. PocketMiner predicts a cryptic pocket "above" the active site that may be targeted for the design of more specific compounds (Fig. 6D). We simulated *PIM2* and found that a cryptic pocket does form in the predicted region (Fig. 6E). Importantly, this pocket has not been previously targeted by a small molecule, but the structures derived from simulation could help lead a drug development campaign to do so. Across the KEGG pathway, there were several other proteins, such as *WNT2*, which lack pockets in their ground state structures but have high cryptic pocket predictions. PocketMiner predicts that *WNT2* has cryptic pockets, and our simulations of this protein captured the formation of a large pocket at the predicted site (Supplementary Fig. 9). Given that *WNT2* lacks any experimental structures, this suggests that applying PocketMiner to AlphaFold structures is a viable strategy for expanding the set of druggable proteins.

## Discussion

We have introduced PocketMiner, a graph neural network for predicting sites of cryptic pockets from folded protein structures. Previous work demonstrated that ligand-binding cryptic pockets could be predicted with reasonable accuracy by training a machine learning algorithm on known examples in the PDB[21]. One drawback of the previous approach is that the prediction is slow to compute since it requires molecular simulations to be computed as an input feature to the algorithm. We hypothesized that we could develop a faster and more accurate algorithm by training a machine learning algorithm on simulation data containing pocket opening events rather than on proteins that are known to have ligand-binding cryptic sites. To test this hypothesis, we trained PocketMiner to predict which residues will form pockets over 2,400 simulations across 35 proteins. This approach led to a model with improved performance (ROC-AUC 0.87 vs. 0.85) and speed (>1000-fold speedup) when compared to CryptoSite after being evaluated against a ground truth set of examples where an experimental structure exists with a ligand bound in a cryptic pocket.

The current work strengthens the case for using molecular dynamics simulations to identify cryptic pockets. Several studies have detailed the discovery of cryptic pockets using molecular simulation[11–16]. In this work, we systematically evaluated how well cryptic pockets are identified by simulation across a large set of known cryptic pockets. To our surprise, we found that most known ligand-binding cryptic pockets can be identified with just 400 ns of aggregate,

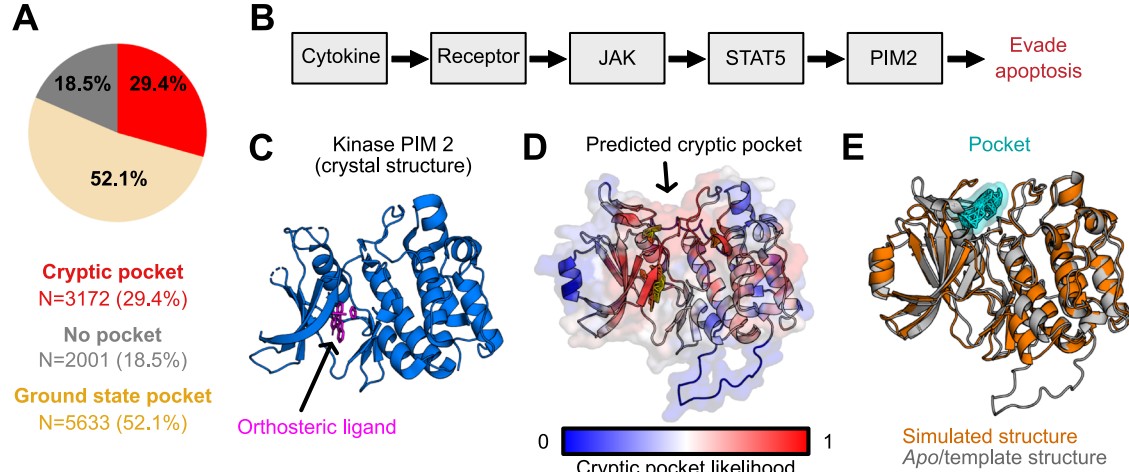

**Fig. 6 | Applying PocketMiner across the human proteome reveals thousands of cryptic pockets. A** Though nearly half of proteins lack a pocket in their native, folded state, the majority of these proteins are predicted to have a cryptic pocket. Pie chart shows the number of human proteins containing pockets in their native structures in wheat; proteins that lack pockets in their native structures but are likely to form cryptic pocket(s) in red; and proteins that lack both ground state and cryptic pockets in gray. **B** Schematic showing proteins involved in the cancer-related Jak/Stat signaling pathway. **C** The crystal structure of the *PIM2* kinase contains an orthosteric binding site but does not reveal any allosteric pockets. **D** PocketMiner predicts a cryptic pocket at an allosteric site. **E** Simulations recapitulate a cryptic pocket predicted by PocketMiner. In simulations, a loop peels back to reveal a cryptic pocket (shown in cyan) at the site pinpointed by PocketMiner. The simulated *PIM2* structure with a cryptic pocket (orange) is overlaid on the *apo PIM2* structure (gray).

unbiased simulation (Fig. 2). Notably, the simulations accurately identify known ligand-binding cryptic pockets without numerous false positives (Supplementary Table 6, Supplementary Fig. 2). Where PocketMiner can be used to identify if a protein has a cryptic pocket, simulations can then be used to sample structural configurations with the pocket open, which enables structure-based drug design. Given that this usually requires only a modest ~400 ns of sampling, this process should be accessible to a wide variety of researchers.

To demonstrate the utility of PocketMiner we applied it across the human genome to identify new cryptic pockets in human proteins. We find over half of the proteins thought to lack pockets are predicted to harbor a cryptic pocket that could render them druggable. Therefore, proteins without obvious pockets in their folded structure should not be overlooked as drug targets. To this point, we highlight *WNT2*, a protein in the Jak/Stat signaling pathway that plays a crucial role in tumorigenesis, lacks an obvious pocket in its folded structure, and is predicted to form a cryptic pocket. Additionally, PocketMiner predicts a cryptic pocket in *PIM2*, a kinase implicated in multiple cancers. In both cases, we used molecular dynamics simulations to verify that those cryptic pockets form (Fig. 6, Supplementary Fig. 9).

With our results indicating that PocketMiner can identify sites of cryptic pocket formation and that simulations can sample open structural states, we propose a pipeline for systematically targeting cryptic pockets. Given a set of proteins implicated in a disease (e.g., SARS-Cov-2 proteome), first apply PocketMiner to these proteins to learn which targets are likely to form cryptic pockets. Then, run MD simulations of those proteins with high PocketMiner scores to sample open structural states. One could even use adaptive sampling to preferentially sample opening events at the site(s) of predicted pockets. These states can be used as templates for structure-based drug design (e.g., molecular docking) to find small molecules that bind at the cryptic pocket. Finally, these hits can be validated with experimentally determined structures or binding assays. Thus, PocketMiner has the potential to become a valuable tool for drug discovery.

## Methods
### Molecular dynamics simulations
**System preparation.** The structures of protein chains selected for simulation were downloaded from RCSB PDB[40]. Structures were

loaded into PyMOL[41] and only the protein chain was saved to generate the simulation input structure. Unresolved internal loops were modeled using SWISS-MODEL[42], with the available crystal structure as a template and the FASTA sequence from the same RCSB PDB entry as the target sequence. Unresolved terminal regions were not modeled.

The structures of protein chains selected for simulation were downloaded from RCSB PDB[40]. Structures were loaded into PyMOL[41] and only the polypeptide chain was saved to generate the simulation input structure. Unresolved internal loops were modeled using SWISS-MODEL[42], with the available crystal structure as a template and the FASTA sequence from the same RCSB PDB entry as the target sequence. Unresolved terminal regions were not modeled.

GROMACS (Gromacs 2020.1)[43] was used to prepare all simulations included in this study. The protein topology was prepared with Amber03[44]. Virtual sites[45] were used to allow for a 4 fs timestep during production MD. It was solvated in a rhombic dodecahedral box of TIP3P water[46] with a minimum distance of 1 nm between the protein and the edges of the box. Na+ and Cl– ions were added to neutralize the net charge of the system and achieve a salt concentration of 0.1 mol/liter.

The protein's potential energy was minimized using the steepest descents algorithm with an initial step size of 0.01 nm. Minimization ran until the maximum force fell below 100 kJ/(mol * nm) or for 500 steps.

The protein was equilibrated for 0.1 ns with a 2 fs time step. All bonds were constrained using the LINCS algorithm[47]. The neighbor list used a Verlet cutoff scheme and a 1.1 nm cutoff radius. A cutoff of 0.9 nm was used for Coulomb and van der Waals interactions. Long-range interactions were treated using the particle mesh Ewald method[48] with a Fourier spacing of 0.12 nm. The velocity-rescaling thermostat[49] was used with the temperature set to 300 K.

**Molecular dynamics simulations.** GROMACS (Gromacs 2020.1 and Gromacs 2021.2) was used to simulate all proteins in this study. A 4 fs timestep was used. The leap-frog algorithm was used for integrating Newton's equations of motion. All covalent bonds involving hydrogen were constrained using LINCS[47]. A cutoff of 0.9 nm was used for Coulomb and van der Waals interactions. Long range interactions were treated using the particle mesh Ewald method[48] with a Fourier spacing

of 0.12 nm. The velocity-rescaling thermostat[49] was used to maintain the system at 310 K and the Parinello-Rahman barostat[50] was used to maintain the system at 1 bar of pressure.

**Adaptive sampling protocols.** Adaptive sampling simulations were performed using the FAST algorithm[26]. FAST takes a single starting structure and an order parameter and uses multiple rounds of parallel unbiased molecular dynamics simulations to search for conformations that maximize or minimize the order parameter. The first round of FAST consists of n parallel trajectories started from a single structure. At the end of each round of FAST, the simulation trajectory frames from all FAST trajectories run thus far are clustered by RMSD and the cluster centers are ranked by the order parameter. The top n cluster centers are then used to start n parallel trajectories for the next round of FAST.

The equilibrated crystal structure of each protein was used as the FAST starting structure in this study. As our ability to generate accurate training labels from simulation depended on the ability of our simulations to sample cryptic pocket opening and we needed an order parameter not dependent on a priori knowledge of pocket location, we used total LIGSITE[30] pocket volume as our order parameter. We clustered using the k-centers algorithm and RMSD between all backbone α-carbons as a distance metric. For each simulation, we determined a cluster radius that resulted in 100–300 clusters and used this same cluster radius for additional rounds of adaptive sampling. We performed either 3, 5, or 7 rounds of adaptive sampling depending on the simulation. For a full list of simulations that were conducted as part of this study, please reference Supplementary Table 4. This study also used pre-existing FAST simulation data from multiple sources, not all of which had used the same FAST parameters[5,12].

## Network architectures

**Geometric vector perceptron-based graph neural networks (GVP-GNNs).** The PocketMiner is an equivariant graph neural network derived from previous work done by Jing et. al[31]. This geometric vector perceptron-based graph neural network (GVP-GNN) was originally designed to learn protein-level representations. In our work, we adapt this model to learn residue-level representations so we can infer each residue's probability of participating in a cryptic pocket.

The GVP-GNN learns a representation of a residue's chemical and topological environment through an information exchange between neighboring residues. Specifically, the input to the network includes node features, which describe a residue's properties, and edge features, which describe relationships between residues. The node features include the following:

- Scalar features {sin, cos} ∘ {φ, ψ, ω}, where φ, ψ, ω are the dihedral angles computed from $C_{i-1}$, $N_i$, $C_{\alpha i}$, $C_i$, and $N_{i+1}$.
- The unit vector in the imputed direction of $C_{\beta i} - C_{\alpha i}$. This is computed by assuming tetrahedral geometry and normalizing.
- Forward and reverse unit vectors.
- A one-hot representation of amino acid identity.

The edge features, which are computed for the nearest 30 neighbors of a residue of interest, include the following:

- The unit vector in the direction of $C\alpha_j - C\alpha_i$.
- The encoding of the distance $||C\alpha_j - C\alpha_i||_2$ in terms of Gaussian radial basis functions.
- A sinusoidal encoding of $j - i$, representing distance along the backbone[31].

Here, we will present an illustrative description of how the network exchanges information between neighboring residues to learn a representation for a residue. We refer the reader to the original work for a mathematically precise description. During each graph propagation step, residue $i$'s representation is updated based on the properties of its 30 nearest neighbors and the spatial and sequence relationship between residue $i$ and each of those neighbors. The neighbor's node features and the edge features between residue $i$ and the neighbor are concatenated such that for a single residue $i$, the network receives 30 vectors. The network transforms these vectors through a "geometric vector perceptron" (see original paper) and then takes the average of the 30 output vectors and adds that to a representation of residue $i$'s self, based on its node features, to get an updated representation of residue $i$. Simultaneously, this happens for all residues in the protein. Residue $i$ (and all others) go through this update procedure $n$ times. The consequence of the iterative updates is that residue $i$'s representation is implicitly affected by its neighbor's neighbors (and so on) because residue $i$'s representation is computed based on its neighbors' representations, which are in turn affected by their own neighbors. Ultimately, each residue gets its own representation that accounts for the topology and chemical environment across the whole protein.

We trained GVP-GNN models using the architecture described above with the following hyperparameters:

- Structural feature node dimensions: 8 vector dimensions, 50 scalar dimensions.
- Structural feature edge dimensions: 1 vector dimension, 32 scalar dimensions.
- GVP hidden dimensions: 16 vector dimensions, 100 scalar dimensions.
- Encoding layers: 4.
- Neighbor list for graph propagation: 30.
- Dropout rate: 0.1.

With minor exceptions, these hyperparameters match those used in the Model Quality Assessment Model in Jing et. al. We tuned these hyperparameters using the optuna software package[51] but found little sensitivity to the choice of these hyperparameters (Supplementary Table 2, see "Model Training – Task 1" for detailed information about hyperparameter selection).

**3D convolutional neural networks (3D-CNNs).** We also tested the ability of a 3D convolutional neural network to predict pocket volume changes in simulations based on starting structures (task 1 in Supplementary Fig. 1). The 3D CNN model we trained is similar to previous models[32], with three 3D convolutional layers with filters of size $3 \times 3 \times 3$ and the following numbers of filters for each of the three layers: 32, 64 128. Each convolutional layer is followed by a max pooling layer and a drop-out layer with drop out probability of 0.7 ("Model Training – Task 1" for detailed information about hyperparameter selection). The last layer is a fully connected layer of size $128 \times 2$.

The 3D CNN model takes a 3D input with four different channels, with one channel per atom type for carbon, oxygen, sulfur, and nitrogen atoms[52]. Given an input protein, first a 3D grid with points 10 Å apart is placed over it with the origin being set to the minimum Cartesian x, y and z coordinates of the structure. Next, every residue in the input structure is assigned to the grid point that is closest to any atom that belongs to it. A 20 Å × 20 Å × 20 Å cube is then defined around the Cβ atom and the cube is oriented such that the plane formed by the N-CA and the C-CA bonds forms the x−y plane and the orthogonal orientation with which the CA- Cβ bond has a positive dot product serves as the positive z-axis. The cube is further divided into 1 Å 3D voxels, within which the presence of the four atom types C, O, N, S is indicated (count of 1 for presence, 0 for absence) to produce the resultant 4 channel 3D grid. Gaussian filters are applied to the discrete counts to approximate atom connectivity and electron delocalization. See fig. S10 for the result of the featurization on a structure from an MD trajectory for the SARS-2-nsp5 monomer protein showing the 3D input for residue 51.

**GVP-GNN and 3D-CNN model training and evaluation (task 1)**
We trained two different architectures for the purpose of predicting pocket volume changes in simulation using starting structures (Task 1).

**Data featurization.** We trained with protein structures from MD simulations with each residue assigned a label depending on whether pockets formed near that residue in the next 40 ns of simulation. Specifically, we calculated pockets using the LIGSITE pocket detection algorithm[30], mapped pockets to residues, and determined the difference for each residue between its assigned pocket volume in the starting structure and the maximum pocket volume for that residue in the simulation. For the LIGSITE calculation implemented in enspara[53], we used a min rank of 7, a grid spacing of 0.7 Å, a probe radius of 1.4 Å, and a minimum cluster size of 3 grid points. To assign pocket volumes to residues, we calculated how many LIGSITE pocket grid points were within 5 Å of each residue. We labeled each residue as a positive example if at some point in the MD simulation (our trajectories were saved out at a rate of 20 ps and we performed computations on every frame) its assigned pocket volume increases by 40 $A^3$ relative to its volume in the starting structure.

**Model evaluation.** We evaluated the performance of GVP-GNNs and 3D-CNNs at predicting the pocket volume changes in simulation based on starting structures using 5-fold cross-validation. We generated 5 folds based on their CAT codes and limited the overlap between folds such that there was no overlap at the level of topology except for the very common 3.40.50 code, which appears 1–3 times in every fold. We performed 5 independent model evaluations using 3 folds for training, 1 fold for validation (i.e., to select a model between training epochs and decide which hyperparameters to select), and 1 fold for testing. For each split, separate models were selected for final testing on the held-out test set based on their PR-AUCs on the validation set.

**GVP-GNN training.** To optimize the performance of the GVP-GNN, we first assessed how different GVP network hyperparameters (i.e., dropout rate and hidden scalar dimension) and learning rate affected performance for a single validation fold. These results are provided in Supplementary Table 2. We found that a low learning rate was associated with improved performance as assessed by PR-AUC. However, the other hyperparameters did not have a substantial effect on GVP-GNN performance in the ranges of interest. As a result, we used the same network hyperparameters as reported in the original GVP paper.

To select an optimal class balancing scheme and batch size, we calculated a PR-AUC separately for each designated validation set for each of the 5 dataset splits. To address the class imbalance between positive and negative residues, we trained models with oversampling of the positive class, undersampling of the negative class, loss weighting, and drawing of balanced 160-residue batches through both undersampling and oversampling. We also trained with and without examples with intermediate values (i.e., residues with pocket volume differences between 10 $A^3$ and 40 $A^3$). To evaluate the relationship between model performance and batch size, we trained with protein-sized batches, 32-residue-sized batches, and 4-residue-sized-batches. We find that model performance was generally better when we used smaller batch sizes but was not significantly affected by the choice of class balancing schemes or inclusion of intermediate examples (Supplementary Table 3). For each split, we selected the GVP-GNN model with the best performance on the validation set (Supplementary Table 3) and report its test set performance (Supplementary Table 1).

All GVP-GNN models were trained on AMD Radeon Vega 20 GPU nodes.

**3D CNN training.** To optimize the performance of the 3D-CNN, we initially compared dropout rates (0.1, 0.3, 0.5, 0.7, 0.9) and learning-rate (1e-2, 1e-3, 1e-4, 1e-5) on a single validation fold and picked the one with the best validation performance (Supplementary Table 4). These values were then kept fixed for the rest of the experiments. Then, for each fold, we did a grid search across the following hyperparameters

and ranges: batch-size (1, 32, 64, 128) and class balancing schemes (None, skew of 1:2, skew of 1:1). Batch size and class-skew both affected model performance and the best hyperparameter set was selected based on a validation fold (Supplementary Table 4). Fig. S11 shows convergence in the Precision-Recall AUC for a 3D CNN model. Finally, for each split, we took the 3D-CNN model with the best performance on a validation set and report its performance on the corresponding test set (Supplementary Table 1).

## Assembly of a novel set of cryptic pockets

We used the steps described below to assemble a novel set of proteins containing cryptic pockets with solved *apo* and *holo* structures. The cryptic pockets in this set have, in many if not all cases, been identified individually by the authors of their respective crystal structures. but have not to our knowledge previously been systematically assembled into one cryptic pocket dataset.

1. We BLAST searched[54,55] the amino acid sequence of every structure in the Binding MOAD ligand binding database (MOAD)[56] against the set of RCSB PDB structures not in MOAD. We removed hits with amino acid sequence identity below 90% (The 90% threshold at this stage was present for programmatic reasons and to ensure inclusion of selenomethionine and selenocysteine mismatches, and subsequent filtering steps ensured that protein structures with 100% sequence identity were ultimately used). We selected each MOAD structure with at least one remaining BLAST hit (other than itself) as a *holo* candidate and selected each PDB BLAST hit as an *apo* candidate.

2. We removed each candidate structure which did not have at least 2.5 Å resolution. This criterion removed NMR structures, which do not have resolutions reported in PDB files. We also excluded multi-conformer X-ray structures (not to be confused with structures which have only a few residues modelled in multiple conformations).

3. We removed each candidate structure which did not have a monomeric biological unit assigned in remark 350 of the PDB file. The author-assigned biological unit was used where available, and the software-assigned biological unit was used elsewhere.

4. We processed each chain of each candidate structure individually, removing the following chains:
   a. We removed chains with gaps longer than 3 residues.
   b. We removed chains with alphabetic residue insertion codes (e.g., 35 A).
   c. We treated *holo* candidate chains without biologically relevant ligands as *apo* candidate chains for the remainder of the analysis. Such chains occurred when only some of the chains in the *holo* candidate crystal structure had occupied binding sites.
   d. We removed *holo* candidate chains in which the *holo* ligand was a polymer.

5. We removed chains with non-canonical residues, except for those containing selenocysteine and selenomethionine, which were treated as cysteine and methionine respectively.

6. We removed *apo* candidate structures with less than 100% sequence identity to their respective *holo* candidate structures, except for mismatches between selenocysteine and cysteine and selenomethionine and methionine (see above). This step did not check for differences between unresolved terminal residues, which can result from variation in the placement of histidine tags. As this step did not use a sequence alignment but instead relied on matching PDB residue numbers, it may have excluded some structures with mismatched PDB residue numbering.

7. We removed *apo* candidate structures with ligands (excluding water, heavy water, sodium, chloride, and potassium) within 5 Å of all MOAD-assigned biologically relevant residues in the *holo* candidate structure (aligned by all-residue Cα RMSD).

8. We selected the *apo* structure with the lowest all-atom Cα RMSD to its *holo* structure for further analysis.
9. We defined the cryptic ligands and ligand-lining residues as follows:
   a. Cryptic ligands are the *holo* ligands with ligand-free *apo* binding sites, plus any associated ions within 3.5 Å which are absent from *apo* (i.e., when ADP and Mg$^{2+}$ bind together, both are treated as cryptic ligands even though Mg$^{2+}$ alone would not be considered a valid cryptic ligand in a *holo* candidate structure).
   b. Ligand-lining residues are residues with any heavy atom within 5 Å of any heavy atom of a cryptic ligand.
10. We ranked and manually filtered the apo-holo candidate pairs as follows:
    a. We ranked all apo-holo pairs in descending order of all-residue Cα RMSD and manually inspected the top 220 proteins* (reaching an RMSD of 0.085 nm). The lowest-ranked protein to survive subsequent rounds of filtering was the 142nd protein on the list (apo PDB ID 6YPK), with an RMSD of 0.124 nm.
    b. We calculated the mean distance between heavy atoms of ligand-lining residues and the apo-holo heavy-atom RMSD of these residues. We sorted the structures in which the ligand-lining residues were closer together in the apo structure than in the holo structure in order of descending RMSD and manually filtered the top 121* (reaching an RMSD of 0.038 nm). The lowest-ranked protein to survive subsequent rounds of filtering was the 86th protein on the list (apo PDB ID 4w51), with an RMSD of 0.123 nm.
    c. *The minimum RMSDs and numbers of proteins to filter manually were determined manually in the course of filtering rather than being derived a priori. We stopped filtering proteins once we concluded that we were encountering only marginal cryptic pocket examples.
11. We removed the following classes of proteins in the course of manual filtering:
    a. monomeric truncation mutants of proteins which oligomerize in their complete forms.
    b. proteins with covalently conjugated *holo* ligands.
    c. proteins with gaps longer than three residues in the resolved structures which had escaped automatic filtering due to anomalies in protein numbering and/or remark 465 of the PDB file.
12. To ensure that our dataset did not contain redundant examples, we calculated the pairwise sequence identities of the proteins selected during manual filtering. Sequence identities were calculated from alignments generated by BioPython[57] using the BLOSUM62 scoring matrix (the same one used by protein BLAST). We removed the protein with the lower cryptic site RMSD when the sequence identity of a pair of proteins exceeds 40%.
13. We removed proteins with sequence identity to a CryptoSite[21], SARS-CoV-2, or 'brick' protein exceeding 40%, except for those listed in the sequence_identity tab of the attached SI spreadsheet, which had very poor structural homologies. See fig. S7 and Supplementary Table 5 for examples of such alignments.
14. To ensure that pockets were larger in the *holo* structure than the *apo* structure, we removed any apo-holo pairs in which the LIGSITE pocket volume assigned to the ligand-lining residues in *apo* (see "Data Featurization" above) was greater than the volume assigned to them in *holo* by at least 20 Å$^3$. As LIGSITE calculates solvent accessible volume rather than van der Waals volume, and volume alone does not indicate steric compatibility, this threshold could not easily be chosen by reference to the volumes of common small molecules. The threshold was therefore assigned manually.

15. Unresolved internal loops were modeled using SWISS-MODEL[42], with the available crystal structure as a template and the FASTA sequence from the same RCSB PDB entry as the target sequence. Unresolved terminal regions were not modeled.

## Identification of residues unlikely to form cryptic pockets

**Proteins extensively crystallized with ligands.** Having reliable examples of protein residues which do not form cryptic pockets is important for measuring the specificity of cryptic pocket prediction methods. However, identifying such residues is difficult because the existence of any single structure in which certain residues do not line cryptic pockets does not prove that those residues would not move to form cryptic pockets in the presence of other ligands. Given the immense size of both chemical space and protein configuration space, it is not possible to conclusively determine that a given protein residue would never form part of a cryptic pocket capable of binding small molecules. However, one can obtain a degree of confidence by using residues in well-studied proteins with many solved *holo* structures which never bind ligands as negative examples. Additional confidence in the stability of these residues can be gained by including *holo* structures of closely related mutants. We collected a set of these residues as described below:

1. We clustered all of the proteins in MOAD to 90% sequence identity using the USEARCH algorithm[58] and ranked the resulting clusters in order of descending size.
2. We BLAST searched the centroid of each USEARCH cluster against the PDB and collected results with sequence identity exceeding 90% and high BLAST scores (ensuring high coverage). The latter criterion eliminated chance and fragmentary alignments which did not represent the complete protein structure. The results included the contents of the USEARCH cluster plus additional structures not in MOAD. Structures containing multiple different sequences were excluded as they are typically hetero-multimeric complexes not consistent with this work's focus on monomeric proteins.
3. We aligned all the sequences of structures in the BLAST output in a multiple sequence alignment (MSA)[59].
4. For each structure in the BLAST results which was in MOAD, we identified all residues within 5 Å of any ligand identified as biologically relevant ('valid') by MOAD. These residues were projected onto the USEARCH cluster centroid using the MSA. The use of MOAD-identified compounds excluded small hydrophilic molecules such as DMSO, glycerol, and ethylene glycol which are miscible in water and are often added as cryoprotectants, as well as small ions. Such substances, when they are present in crystal structures, may sit at any point on the protein surface and are unlikely to occupy larger pockets in the hydrophobic interior of the protein.
5. The residues of the USEARCH cluster centroid which were not found within 5 Å of any valid MOAD ligand were taken as candidate negative examples.
6. As this study focuses on predicting cryptic pockets in monomeric proteins, we removed clusters which did not have more than 50% of the structures labeled as monomeric. The oligomerization state of each PDB BLAST hit was determined from remark 350 of the PDB file, using the author-assigned state if available and the software-assigned state otherwise.
7. Clusters were ranked by the number of unique MOAD ligands they contained.
8. The top 12 remaining clusters and lysozyme, which ranked 22nd in number of unique MOAD ligands but has been extensively studied and is known to be relatively thermostable, were selected for further analysis.
9. Clusters with centroids having more than 40% sequence identity to proteins in the training set were removed, eliminating two clusters.

10. For each of the remaining clusters, the *apo* structures identified from the PDB BLAST were ranked by their sequence homology to the consensus sequence for that USEARCH cluster, and the *apo* structure with the greatest homology was selected for use in the simulations described below. In some cases, *holo* structures not indexed in MOAD were eliminated to identify a suitable *apo* structure.

**Highly stable and/or highly rigid proteins.** To identify additional examples of residues which are unlikely to form cryptic pockets, we searched for proteins which had previously been studied as examples of extremely stable and/or rigid proteins. These proteins have been characterized by methods such as nuclear magnetic resonance (NMR) and circular dichroism (CD) spectroscopy which do not rely on sampling the chemical space of possible cryptic ligands. NMR measurements suggest that ubiquitin, the third IGG binding domain from streptococcal protein G, and flavodoxin (PDB IDs 4HJK, 1IGD, and 1OFV respectively) have minimal internal motion[60–62]. γ-B Crystallin (PDB ID 1AMM) is believed to have a highly rigid structure based on tryptophan fluorescence measurements of a γ-D crystallin with which it shares its fold and all four relevant tryptophan residues[63]. Hydrogen deuterium exchange (HDX) experiments demonstrate extremely slow solvent exposure of internal residues of alpha-lytic protease (PDB ID 2ALP), severely restricting the set of residues which could possibly form pockets on biologically relevant timescales[64]. Crambin, a designed three helix bundle, Top7, and a designed TIM barrel (PDB IDs 2FD7, 4TQL, 1QYS, and 5BVL respectively) lacked observables providing similarly direct evidence of ground state rigidity and were chosen on the basis of thermodynamic stability on the assumption that many such proteins would also have deep ground state energy wells which would disfavor cryptic pocket formation. Crambin was characterized by thermal denaturation measured by $^1$H-NMR[65], while the other three were characterized by thermal and chemical denaturation measured by CD spectroscopy[66–68] (slow unfolding kinetics were reported for the designed three-helix bundle well). However, as CD spectroscopy itself only measures secondary structure content, such assays do not rule out the formation of cryptic pockets involving other types of protein motion at temperatures well below the melting point. Six of the above proteins are naturally occurring, and the other three (the three-helix bundle, Top7, and the TIM barrel) are engineered proteins, one of which (Top7) has a novel fold. Two of the natural proteins (alpha-lytic protease and flavodoxin) possess non-cryptic ligand-binding pockets while the remainder are believed to lack pockets. The residues within 5 Å of those ligands were not used as negatives, while all other residues of the resulting 9 proteins were taken as candidate negative examples. To build further confidence in negative labels, we also conducted simulations of these proteins and tracked cryptic pocket opening in those simulations.

**Simulations.** Molecular dynamics simulations were performed to search for additional cryptic pockets not yet identified experimentally and to ensure a high degree of confidence in the negative labels. We ran simulations of the 9 hyper-rigid proteins and the 10 exemplars of proteins with extensive *holo* crystal structures. Candidate negative residues adjacent to pockets which formed in simulation were eliminated. Specifically, we eliminated any residues that had an assigned LIGSITE pocket volume greater than 20 Å$^3$. This threshold was determined by finding the residue-level LIGSITE pocket volumes in 13 *holo* structures from the CryptoSite dataset and selecting a value at the 10$^{th}$ percentile of the distribution. The remaining negative residues that both (a) do not bind any ligands in extensive *holo* crystal structures and (b) were not adjacent to a cryptic pocket in simulation were used as negative true labels in the validation and test sets.

## PocketMiner training and evaluation (Task 2)
PocketMiner was trained with simulation data to predict ligand-binding cryptic pockets in experimental structures (Task 2).

**Data featurization.** Like in the previous prediction task, we trained with protein structures from MD simulations with each residue assigned a label depending on whether pockets formed at that residue in the next 40 ns of simulation. Because there are multiple ways to calculate pockets and assign them to nearby residues, we trained with several labeling schemes. We used the LIGSITE pocket detection algorithm with the hyperparameters described above as well as the fpocket detection tool with default parameters[35]. We assigned LIGSITE volumes to nearby residues in two different ways: (1) by assigning all pocket grid points within 5 Å of each residue and (2) assigning each pocket grid point to the nearest residue.

The fpocket algorithm identifies cavities in the protein. Each pocket is a composite of both alpha spheres and the associated residues within that pocket. The fpocket algorithm implements a ranking function to estimate the druggability of the concavities in a protein. To generate residue-level labels, we iterated through all pockets in the protein and assigned each residue either the maximum score of its nearby pocket(s) or a value of 0 if the residue did not participate in a pocket.

Next, we found the maximum increase in residue-level pocket volumes or druggability scores over the course of the 40 ns simulation (for fpocket druggability scores, we also trained with labels based on the maximum drug score in the simulation rather than the maximum increase). To find the maximum increase over a 40 ns window, we determined pocket volumes for every structure spaced 20 ps apart and fpocket druggability scores for every structure spaced 100 ps apart. Finally, we binarized the assigned LIGSITE pocket volumes at different thresholds that were determined based on an analysis of pocket formation at known cryptic sites (e.g., 20, 30, and 40 Å$^3$ for the first assignment procedure). Fpocket druggability scores were not binarized since they are already between 0 and 1.

**GVP-GNN training.** We trained with the same hyperparameters as described in the Methods section entitled "GVP-GNN and 3D-CNN Model Training and Evaluation" on AMD Radeon Vega 20 nodes. Models trained with different residue labels (LIGSITE pocket volume changes across different binarization thresholds vs. fpocket druggability scores), batch sizes (1 protein batches vs. 4 residue batches), and class balancing schemes (no balancing vs. constant size balanced draws of 640 residues vs. undersampling negatives) were compared using a validation set of experimental structures including hyper rigid proteins, proteins with extensive ligand-bound crystal structures, and examples of cryptic pockets. We found that a model trained for 20 epochs with LIGSITE-derived labels and refined for 1 epoch with labels derived from fpocket druggability scores had the best validation AUC among GVP-GNNs that we trained. This model, dubbed PocketMiner, was trained with 4 residue batches where labels were based on finding the difference in LIGSITE pocket volumes assigned by mapping each pocket grid point to its nearest residue in the first phase of training. Residue labels were binarized using a threshold of 20 Å$^3$. In the second phase of training, the same residues were relabeled using the maximum fpocket druggability score assigned to each residue based on a 3 A cutoff.

**Model evaluation.** Following the selection of a single model, we evaluated the performance of PocketMiner using a held-out test set. This test set included 24 *apo* structures with ligand-binding cryptic pockets, 4 hyper-rigid proteins, and 7 proteins that were the subjects of extensive ligand screening (see Methods section entitled "Identification of non-cryptic-pocket-forming residues from extensively crystallized proteins and hyper-rigid proteins"). None of the protein structures in our test set had more than 60% sequence similarity to our training proteins, but the proteins with >40% sequence identity had

very poor structural homology (Supplementary Fig. 7, Supplementary Table 5).

## Applying PocketMiner to the human proteome
To obtain a protein for each gene in the human genome, we used the UNIPROT human reference proteome (UP000005640)[69]. For each protein, we checked the Protein Data Bank for an experimentally determined structure that is monomeric, has resolution <2.5 Å, and is between 50 and 1000 amino acid residues in length. If there was more than one such entry, we chose the longest version of the protein. If there was no experimentally determined structure, we used the AlphaFold predicted structure[19]. We only used the AlphaFold predicted structure if it had no stretches of more than 25 residues with a low confidence prediction (i.e., <70 pLDDT confidence score), excluding the N- and C- terminal segments of the protein. We chopped low confidence N- and C-terminal segments from the protein structure because our validation and test sets did not contain disordered structural elements. In the case of extremely large proteins, AlphaFold breaks the prediction into fragments. We only considered the first (N-terminal) fragment for these proteins.

We classified the remaining 10,806 structures into one of three categories: those proteins containing "ground state pockets"; those proteins containing "cryptic pockets" only; and proteins with neither ground state nor cryptic pockets ("no pocket" in Fig. 6). For each structure, we calculated PocketMiner's prediction across all residues and calculated the size of the largest pocket in the structure using the enspara implementation of the LIGSITE algorithm. For LIGSITE, we used a min rank of 7 and a minimum cluster size of 3 grid points. We considered proteins to have a "ground state pocket" if they contained a LIGSITE pocket with volume over 30 Å[3] (Fig. 6). We then determined if all the remaining proteins formed cryptic pockets using PocketMiner's predictions. Specifically, we looped over all residues in the protein and took the average of that residue's prediction and its 10 neighboring residues to define a cryptic pocket hotspot. If a protein had a hotspot with a score over 0.7, we considered it to have a cryptic pocket. This threshold was based on applying this score to the proteins in our test set where proteins with ligand-binding cryptic pockets were considered positive examples and hyper rigid proteins were considered negative examples. At a threshold of 0.7, we achieve an accuracy of 0.90 with 3/4 negative examples correctly identified and 21/24 positive examples correctly identified.

Finally, we manually inspected a random sampling of proteins in the KEGG pathway "Pathways in Cancer"[38]. During this manual inspection, we identified proteins (*PIM2* and *WNT2*) with high PocketMiner predictions on regions of the protein that lacked a pocket. Then, we simulated *PIM2* and *WNT2* using the same procedure as all other simulations described. We then analyzed the simulations for pocket opening using LIGSITE as described above.

## Reporting summary
Further information on research design is available in the Nature Portfolio Reporting Summary linked to this article.

## Data availability
All data supporting the findings of this study are available within this study, the code repository, and the supplementary information files or available on request. The dataset of cryptic pockets that was generated as part of this work can be found in the Supplementary Data 1. The following PDB codes were used in the figures: 1JWP, 1PZO, 1MY0, 1N0T, 2W9T, 2W9S, 6E5D, 6E5F, 1Y1A, 4P0I, 5OTA, 4TQL. Source data are provided with this paper.

## Code availability
The PocketMiner web interface is available at https://pocketminer.azurewebsites.net/. The PocketMiner codebase is freely available on Github (https://github.com/Mickdub/gvp/tree/pocket_pred) and Zenodo[70]. Code implementing the 3D-CNN in PyTorch as well as checkpoint files containing the best 3D-CNN model for each task 1 dataset split are available at https://github.com/meghana-kshirsagar/3DCNN_protein_structures/tree/main/models.

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

## Acknowledgements

We would like to thank Microsoft AI for Health for an Azure grant as well as Fabian Salamo, Marcelo Duarte, Pedro Costa and Anthony Citron for help in developing the PocketMiner web interface. We are grateful to Neha Vithani for proposing the name PocketMiner and Bowen Jing for help in using the geometric vector perceptron codebase. We would like to thank AMD for the donation of critical hardware and support resources from its HPC Fund that enabled the computations for this work. We would like to acknowledge the Folding@home community for its support. AM was supported by the National Institutes of Health F30 Fellowship (1F30HL162431-01A1). JML was supported by the National Science Foundation via grant number DGE-2139839. MDW was supported by a MolSSI COVID-19 seed software fellowship. This work was funded by NSF CAREER Award MCB-1552471 (GRB) and NIH grants R01 GM124007 (GRB) and RF1AG067194 (GRB). G.R.B holds a Packard Fellowship for Science and Engineering from The David & Lucile Packard Foundation.

## Author contributions

AM, MDW, JB, MK, FO, and JML conceptualized and performed the analyses for this study. AM performed molecular dynamics simulations and created training labels. JB curated the cryptic pocket dataset that was used for model evaluation. AM, MDW, and JB created the figures for the manuscript. AM and MDW drafted the manuscript. AM, MDW, FO, MK conceptualized, wrote, and tested software. JLF provided edits to the manuscript and acquired funding. GRB proposed the initial project idea, provided edits to the manuscript, and acquired funding.

## Competing interests

GRB is a co-founder and equity holder in Decrypt Biomedicine. The remaining authors declare no competing interests.
