## [Peer review file · Nature Communications]

REVIEWER COMMENTS

Reviewer #1 (Remarks to the Author):

Overall this is an excellent paper tackling a problem of immense relevance, where the authors shows that their method is clearly outperforming the competition at the defined task. I believe it is very worthy of publication, however I do have several concerns that I would like to see discussed/addressed in a revised version. Most of them should be addressable just through a simple discussion.

1. I don't entirely understand why authors are only using ROC AUC as their metric for this problem when it can be misleading with class imbalanced data (the majority of atoms are not in sites). They later say only ~10% of the data is in the positive class and start using PR AUC as a metric which should be more meaningful in this case
2. It's strange to treat LIGSITE as a ground truth when it had ~25% accuracy in this assessment <https://www.sciencedirect.com/science/article/pii/S0969212611001079> Is the assumption here that LIGSITE does a very good job of identifying cavities but a very poor job of ranking them and this new method only cares about spatial opening and not "ligandability"? Is there a study that shows LIGSITE gets pocket locations right but can't rank them correctly? This was later clarified to be only volumes not scores. Is there study showing the volumes are good?
3. Can we make claims about "druggability" if we are only talking about pocket opening and not known binders? I would think chemical as well as geometric features are needed to say anything about the likelihood that a drug will bind with sufficient affinity as opposed to the likelihood that we just have existence of a cavity. The test set is done without reference to actual binding events, just pocket opening. I think the claims should be refined to show that this method predicts where pockets open not where drugs bind - OR, provide clear, quantitative justifications from literature as to why pocket opening implies high drug binding success
4. Fpocket scores are used to assess druggability but fpocket has just as bad performance on the Chen assessment (~25%). Are these really trustworthy scores?
5. The data leakage criterion for the test split and the comparison with Cryptosite seem rigorous however it must be made very clear that the way cryptosite is trained has a reference to ligandability

and this no longer does. The task has slightly changed because Cryptosite tests on known cryptic sites which a priori have a drug bound while this tests on potential binding sites^[SEP]

6. It's a bit strange to tell us the test set accuracy before finishing describing the test set.^[SEP]

7. The training procedure was confusing with how fpocket was used for refinement until reading the methods section. Please rewrite more clearly.^[SEP]

Reviewer #2 (Remarks to the Author):

Really exciting work on cryptic pocket prediction! This research provides an approach that delivers cryptic pocket predictions with a higher efficiency than the state-of-art Cryptosite, and a potential approach for allosteric binding pocket predictions as well. The researchers present a unique approach to utilize AI models in predicting protein druggability, potentially facilitating real world drug discovery efforts and rendering previously intractable protein targets as valid subjects for medicinal chemistry. The work substantially supports the claims made, which are limited and well-defined. Results are presented in a clear and convincing fashion, with well thought-out methodology.

Two minor questions here:

1. Could you provide a detailed list of hyper-parameters you used on GVP GNN and comparisons of their performances? I think this would be useful information for people to adapt this method for other similar research, and for the sake of reproducibility.
2. Did you perform hyper parameters tuning on the 3D GNN ? If not, please explain why you chose such a high drop out rate (0.7) which might interfere with the model performance. If you did, please use the hyperparameter tuning results to support this drop out rate.

REVIEWER COMMENTS

Reviewer #1 (Remarks to the Author):

Overall this is an excellent paper tackling a problem of immense relevance, where the authors shows that their method is clearly outperforming the competition at the defined task. I believe it is very worthy of publication, however I do have several concerns that I would like to see discussed/addressed in a revised version. Most of them should be addressable just through a simple discussion.

Thank you for the positive feedback. We have followed the reviewer's advice and made minor modifications to address the comments below.

1. I don't entirely understand why authors are only using ROC AUC as their metric for this problem when it can be misleading with class imbalanced data (the majority of atoms are not in sites). They later say only ~10% of the data is in the positive class and start using PR AUC as a metric which should be more meaningful in this case

Thank you for this insight. It is correct that for our first prediction task (predicting pocket volume changes in simulations based on starting structures), the majority of the data (~90%) is in the negative class as the reviewer observes. In our revised manuscript under the section "Graph neural networks accurately predict the time evolution of pockets in simulation," we have placed emphasis on the PR-AUC (though we report both because many readers are probably more familiar with ROC-AUC). We have also rearranged Fig. 3 to emphasize the PR-AUC curve. In our second prediction task (predicting ligand-binding cryptic pockets from experimental structures), our test set is reasonably balanced (~30% positive), which is why we report a ROC-AUC.

2. It's strange to treat LIGSITE as a ground truth when it had ~25% accuracy in this assessment <https://www.sciencedirect.com/science/article/pii/S0969212611001079> Is the assumption here that LIGSITE does a very good job of identifying cavities but a very poor job of ranking them and this new method only cares about spatial opening and not "ligandability"? Is there a study that shows LIGSITE gets pocket locations right but can't rank them correctly? This was later clarified to be only volumes not scores. Is there study showing the volumes are good?

Thank you for pointing us to this reference and raising this concern. The reviewer is correct in his or her suggestion that the success of our method does not depend on LIGSITE's ability to rank cavities as was assessed in the Chen et al. manuscript. Our training labels are derived from *changes* in the LIGSITE pocket volume in the vicinity of a residue over the course of MD simulations. We have found that across 12 ligand-binding cryptic pocket examples that the change in local pocket volume over the course of simulations accurately predicts which residues participate in ligand-binding cryptic pocket (Table S3). To address this concern explicitly, we have added the following paragraph to our Results section which discusses the findings of Chen et al.:

Though previous studies have found that LIGSITE and fpocket can struggle to correctly rank which pockets bind ligands⁴⁸, our labels did not consider the ranking between pockets in a single structure. Instead, our labels were based on the change in the LIGSITE pocket volume or the maximum fpocket druggability score in the vicinity of a residue over the course of MD simulations. We then evaluated if models trained using these labels could distinguish residues at ligand-binding cryptic sites from residues that do not form cryptic pockets.

3. Can we make claims about “druggability” if we are only talking about pocket opening and not known binders? I would think chemical as well as geometric features are needed to say anything about the likelihood that a drug will bind with sufficient affinity as opposed to the likelihood that we just have existence of a cavity. The test set is done without reference to actual binding events, just pocket opening. I think the claims should be refined to show that this method predicts where pockets open not where drugs bind - OR, provide clear, quantitative justifications from literature as to why pocket opening implies high drug binding success

Thank you for raising this point. The reviewer is correct that the first prediction task (predicting pocket volume changes in simulations based on starting structures) does not reference actual binding events. As a result, strong performance on task 1 does not necessarily indicate that the model can evaluate if a protein structure is druggable. On the other hand, our second prediction task requires the model to predict where ligands will bind cryptic pockets in experimental structures. Our test set for this second task is derived from newly identified cryptic pocket examples in the PDB. We clarify these two prediction tasks in a new supplemental figure that is shown below:

Task 1: Predict pocket volume changes in simulation

x5 for 5-fold cross validation

Task 2: Predict ligand-binding cryptic pockets in experimental structures

Nonetheless, we have followed the reviewer’s advice and removed any claims about protein druggability as other factors beyond the presence of cryptic pockets may affect whether a protein is a viable drug target.

4. Fpocket scores are used to assess druggability but fpocket has just as bad performance on the Chen assessment (~25%). Are these really trustworthy scores?

This is an astute comment from the reviewer. Fpocket's ranking of identified pockets via 'druggability scores' rarely assigns high ranks to known ligand-binding sites. Fortunately, our labeling scheme does not rely on Fpocket's ranking among pockets. Instead, residue-level labels are derived based on the highest scoring pocket within the local vicinity of a residue. Furthermore, when we compared the performance of different labeling schemes on a validation set of experimental structures, we found that 'refining' models with Fpocket-derived labels improved performance (Table S1). We speculate that our strong performance on the second prediction task (predicting ligand-binding cryptic pockets from experimental structures) results from Fpocket's ability to annotate relevant pockets, even if it cannot usually rank among pockets identified in a structure.

5. The data leakage criterion for the test split and the comparison with Cryptosite seem rigorous however it must be made very clear that the way cryptosite is trained has a reference to ligandability and this no longer does. The task has slightly changed because Cryptosite tests on known cryptic sites which a priori have a drug bound while this tests on potential binding sites.

We thank the reviewer for recognizing that our data leakage criterion and test split were rigorous. While the first prediction task (predicting pocket volume changes in simulations based on starting structures) does not test on known cryptic sites, the second prediction task does. The new supplemental figure is meant to clarify this distinction.

6. It's a bit strange to tell us the test set accuracy before finishing describing the test set.

This is a reasonable comment. The focus on two separate prediction tasks may have created confusion in the original manuscript. We anticipate that the addition of Fig. S1 and the following changes in the section headers will make the revised manuscript clearer:

- Graph neural networks accurately predict the time evolution of pockets in simulation
- PocketMiner accurately predicts *ligand-binding* cryptic pockets from ligand-free crystal structures

7. The training procedure was confusing with how fpocket was used for refinement until reading the methods section. Please rewrite more clearly.

Thank you for this observation. We have added the following text to the main results section to clarify the training procedure:

As a result, we took all proteins from our previous training set and generated training labels based on local changes in LIGSITE pocket volume and fpocket druggability scores⁴⁷ (see Methods). We decided to use fpocket labeling schemes since druggability scores consider not only the geometry of a pocket but also the chemical environment of a pocket. We trained GVP-GNNs using LIGSITE-derived labels only, fpocket-derived labels only, and a combination of

both labels (i.e., where a network is trained for several epochs using one labeling scheme and then switches to learning from the other labels for several more epochs).

We find that our final model, referred to as PocketMiner, achieves very good performance at discriminating residues that form cryptic pockets from those that do not (ROC AUC: 0.87). **PocketMiner was trained for 20 epochs with LIGSITE-derived labels and refined for 1 epoch with labels derived from fpocket druggability scores.**

Reviewer #2 (Remarks to the Author):

Really exciting work on cryptic pocket prediction! This research provides an approach that delivers cryptic pocket predictions with a higher efficiency than the state-of-art Cryptosite, and a potential approach for allosteric binding pocket predictions as well. The researchers present a unique approach to utilize AI models in predicting protein druggability, potentially facilitating real world drug discovery efforts and rendering previously intractable protein targets as valid subjects for medicinal chemistry. The work substantially supports the claims made, which are limited and well-defined. Results are presented in a clear and convincing fashion, with well thought-out methodology.

Two minor questions here:

1. Could you provide a detailed list of hyper-parameters you used on GVP GNN and comparisons of their performances? I think this would be useful information for people to adapt this method for other similar research, and for the sake of reproducibility.

We performed hyperparameter scans using the optuna python package while we were training models on the first prediction task (predicting pocket volume changes in simulations based on starting structures). We have added a new supplementary table which summarizes the results of this hyperparameter scan (Table S7). We found limited sensitivity to the choice of hyperparameters we explored. The new supplementary table is shown here for convenience:

Learning Rate	Dropout	GVP hidden scalar dimension	Validation Loss
6.1E-05	0.25	75	0.521
7.6E-04	0.12	75	0.497
6.5E-05	0.16	100	0.519
1.4E-04	0.10	75	0.546
1.7E-04	0.10	100	0.557
6.1E-04	0.09	75	0.524
6.4E-05	0.07	50	0.555
8.0E-05	0.11	100	0.535
3.0E-04	0.09	50	0.583
7.9E-05	0.14	75	0.528

2. Did you perform hyper parameters tuning on the 3D GNN ? If not, please explain why you chose such a high drop out rate (0.7) which might interfere with the model performance. If you did, please use the hyperparameter tuning results to support this drop out rate.

To optimize the performance of the 3D-CNN, we tuned the following hyper-parameters in the mentioned ranges: learning-rate (1e-2, 1e-3, 1e-4, 1e-5), batch-size (1, 4, 16, 32, 64) and dropout rate (0.1, 0.3, 0.5, 0.7, 0.9) during 5-fold cross validation for task 1 (predicting pocket volume changes in simulation based on starting structure). We found the optimal values to be 1e-4, 32 and 0.7 respectively. Overall, we find that a low learning rate, a high dropout rate and a batch-size of 32 supported a monotonically reducing training loss and the lowest validation loss. This text has been added to the methods section.

REVIEWER COMMENTS

Reviewer #1 (Remarks to the Author):

Thank you for addressing all my comments! Excellent work and I happily recommend publication as is.

Reviewer #2 (Remarks to the Author):

Still not convinced that GVP GNN is better than the 3D GNN model.

The authors should perform the same split on the dataset (as identical train, validation and test sets) and use these same sets to train and validate the GVP GNN and 3D GNN models. 5-fold Cross-validation and a hold-out test set (a subset of data that is not used in cross-validation) are recommended. The average validation set results should be used to select the best hyperparameters for each model. Then the test set should be used to compare the performance of GVP GNN and 3D GNN models using the best hyperparameters identified. All the hyperparameters results (cross-validation and test set results) should be listed explicitly in the supplement information.

Also, the checkpoints of the best GVP GNN and 3D GNN models should be provided to readers, in addition to the code base, for reproducibility reasons.

Reviewer #2 (Remarks to the Author):

Still not convinced that GVP GNN is better than the 3D GNN model.

The authors should perform the same split on the dataset (as identical train, validation and test sets) and use these same sets to train and validate the GVP GNN and 3D GNN models. 5-fold Cross-validation and a hold-out test set (a subset of data that is not used in cross-validation) are recommended. The average validation set results should be used to select the best hyperparameters for each model. Then the test set should be used to compare the performance of GVP GNN and 3D GNN models using the best hyperparameters identified. All the hyperparameters results (cross-validation and test set results) should be listed explicitly in the supplement information.

Also, the checkpoints of the best GVP GNN and 3D GNN models should be provided to readers, in addition to the code base, for reproducibility reasons.

Response:

We would like to thank the reviewer for helping us clarify the comparison between the 3D-CNN and GVP-GNN models. We have made several changes and additions based on the reviewer's feedback.

In the main text, we clarify that the 3D-CNN and GVP-GNN models give similar performance, as follows:

We find that both the GVP-GNN and the 3D-CNN learn to accurately classify whether a given residue will form a cryptic pocket in simulation based on a starting structure. We split the 37 protein simulation datasets into 5 folds by protein and used 5-fold cross-validation to measure the performance of both architecture types. For each split, we separately assessed how choices in model hyperparameters (e.g., dropout rate) and training set up (e.g., class balancing scheme) affected performance on 1 validation fold (3 folds were used for training). The GVP-GNN and 3D-CNN that performed best on each validation fold (Table S2-4) for a certain split was then assessed using a held-out test fold (Table S1). Across the 5 splits, the best GVP-GNN model achieves an average test PR-AUC of 0.44 ± 0.12 (average ROC-AUC: 0.83 ± 0.04 ; Fig. 4B-C). The 3D-CNN performs similarly (PR-AUC: 0.41 ± 0.05 ; ROC-AUC: 0.79 ± 0.02).

As suggested by the reviewer, we used exactly the same dataset split to compare the performance of the 3D-CNN with the GVP-GNN. Instead of using a single held-out test set, we used 5-fold cross validation. As suggested by the reviewer, we used a validation set to select optimal hyperparameters for each architecture type separately before comparing their performance on held-out test sets. We did not use nested cross-validation as suggested because the computational expense with deep learning models was prohibitive. Table S1 summarizes the comparison of the 3D-CNN and GVP-GNN on the test sets. It is also provided below for convenience:

Split	GVP-GNN Test PR-AUC	3D-CNN Test PR-AUC
0	0.32	0.36

1	0.40	0.36
2	0.48	0.45
3	0.64	0.49
4	0.38	0.39

In this revision, we have explicitly provided the details of the hyperparameter search in the supplement (Tables S2-S4). These are summarized here as well:

To optimize the performance of the GVP-GNN, we first assessed how different GVP network hyperparameters (i.e., dropout rate and hidden scalar dimension) and learning rate affected performance for a single validation fold. These results are provided in Table S2. We found that a low learning rate was associated with improved performance as assessed by PR-AUC. However, the other hyperparameters did not have a substantial effect on GVP-GNN performance in the ranges of interest. As a result, we used the same network hyperparameters as reported in the original GVP paper. To select an optimal class balancing scheme and batch size, we calculated a PR-AUC separately for each designated validation set for each of the 5 dataset splits. For each split, we selected the GVP-GNN model with the best performance on the validation set (Table S3) and report its test set performance (Table S1).

To optimize the performance of the 3D-CNN, we initially compared dropout rates (0.1, 0.3, 0.5, 0.7, 0.9) and learning-rate ($1e-2$, $1e-3$, $1e-4$, $1e-5$) on a single validation fold and picked the one with the best validation performance (Table S4). These values were then kept fixed for the rest of the experiments. Then, for each fold, we did a grid search across the following hyperparameters and ranges: batch-size (1, 32, 64, 128) and class balancing schemes (None, skew of 1:2, skew of 1:1). Batch size and class-skew both affected model performance and the best hyperparameter set was selected based on a validation fold (Table S4). Finally, for each split, we took the 3D-CNN model with the best performance on a validation set and report its performance on the corresponding test set (Table S1).

Finally, we have shared checkpoint files containing the best 3D-CNN model for each dataset split. These are available at https://github.com/meghana-kshirsagar/3DCNN_protein_structures/tree/main/models. We have also added checkpoint files for the best performing GVP-GNN for each dataset split to our PocketMiner Github. These files will allow future readers to replicate our comparison of the GVP-GNN with the 3D-CNN.